

# Long-term participation in collaborative fisheries research improves angler opinions on marine protected areas

Erica T. Mason[1], Allison N. Kellum[1], Jennifer A. Chiu[2], Grant T. Waltz[3], Samantha Murray[1], Dean E. Wendt[3], Richard M. Starr[2] and Brice X. Semmens[1]

[1] Scripps Institution of Oceanography, University of California, San Diego, La Jolla, CA, USA
[2] Fisheries and Conservation Biology, Moss Landing Marine Laboratories, Moss Landing, CA, USA
[3] Department of Biological Sciences, California Polytechnic State University - San Luis Obispo, San Luis Obispo, CA, USA

Corresponding author
Erica T. Mason, etmason@ucsd.edu

## ABSTRACT

Recent marine spatial planning efforts, including the management and monitoring of marine protected areas (MPAs), increasingly focus on the importance of stakeholder engagement. For nearly 15 years, the California Collaborative Fisheries Research Program (CCFRP) has partnered volunteer anglers with researchers, the fishing industry, and resource managers to monitor groundfishes in California's network of MPAs. While the program has succeeded in generating sustained biological observations, we know little about volunteer angler demography or the impact of participation on their perceptions and opinions on fisheries data or MPAs. In this study we surveyed CCFRP volunteers to learn about (a) volunteer angler demographics and attitudes toward groundfish management and stock health, (b) volunteer angler motivations for joining and staying in the program, and (c) whether participation in the program influenced volunteer angler opinions on the quality of fisheries data used in resource management and the establishment of MPAs in California. CCFRP volunteers were older and had higher fishing avidity than average within the California recreational angling community. Many self-identified as more conservation-minded than their peers in the recreational fishing community and had positive views of California groundfish management and stock health. Participation in science and giving back to fisheries resources were major motivating factors in their decision to become and remain CCFRP volunteers. Angler opinions toward MPAs were more positive after volunteering with CCFRP. Those who had volunteered for seven or more years with CCFRP were more likely than not to gain a positive opinion of MPAs. Our survey results provide evidence that long-term engagement of stakeholders in collaborative research positively influences stakeholder opinions regarding marine resource management, and highlights CCFRP's success in engaging citizen science stakeholders in collaborative fisheries research.

## INTRODUCTION

Stakeholder engagement is an important part of marine resource protection and management (*Pomeroy & Douvere, 2008*). Benefits of such engagement include the incorporation of local knowledge into policy, and the potential to build stakeholder trust in management decisions (*Yochum, Starr & Wendt, 2011*). California's Marine Life Protection Act (MLPA) of 1999 (Fish and Game Code § 2850–2863) directed the state to redesign California's marine protected areas (MPAs) to function as a network and increase protection of the state's marine habitats, wildlife, and cultural sites. Establishment of the statewide MPA network took 13 years and involved considerable effort to engage stakeholders (*Gleason et al., 2010*). Multiple assessments of the MLPA stakeholder process found that it effectively achieved broad participation of resource users and interest groups across the state (*Fox et al., 2013*; *Gleason et al., 2013*; *Kirlin et al., 2013*). Since implementing the MPA network, the state (i.e., California Department of Fish and Wildlife (CDFW), California Ocean Protection Council (OPC), and California Fish and Game Commission (FGC)), in partnership with California Ocean Science Trust (OST), has supported public engagement through continued scientific monitoring of fish populations inside and outside MPA boundaries (*OST, CDFW & OPC, 2012*, *2017*).

The California Collaborative Fisheries Research Program (CCFRP) was created in 2006 by a coalition of scientists, resource managers, commercial passenger fishing vessel (CPFV) operators, and recreational anglers (*Wendt & Starr, 2009*). The program uses a hook-and-line sampling design to monitor the abundance and size of groundfish species (e.g., rockfish, flatfish, roundfish and skates and rays); data are used to make long-term comparisons of species diversity, catch rates, and length-frequency distributions within and among paired MPA and reference sites (sites similar in character and location to the MPA, but without the protections afforded the MPA). Between 2007 and 2016, CCFRP annually surveyed four sets of MPAs along the central coast including Año Nuevo State Marine Reserve (SMR), Point Lobos SMR, Piedras Blancas SMR, and Point Buchon SMR (Fig. 1). These MPAs were implemented in 2007 and are located within the Central Coast MLPA study region that extends from Pigeon Point to Point Conception, CA.

California Collaborative Fisheries Research Program's volunteer saltwater recreational anglers are central to the program's standardized hook-and-line survey. Once a fish is caught by a volunteer angler, it is processed by scientists who identify, measure, tag (externally), and release the catch (*Wendt & Starr, 2009*). Volunteering requires a commitment to early mornings and near constant fishing effort without the benefit of keeping the day's catch. Nevertheless, volunteering gives participants the unique opportunity to fish in MPAs designated as no-take zones where fishing is otherwise prohibited. It also offers volunteer anglers the opportunity to interact with the science crew and make their own observations on the similarities and differences between the catches in MPAs and reference areas. This participation, and the direct observation of the scientific process, is thought to instill a sense of ownership and trust among participants in the data being collected (*Yochum, Starr & Wendt, 2011*).

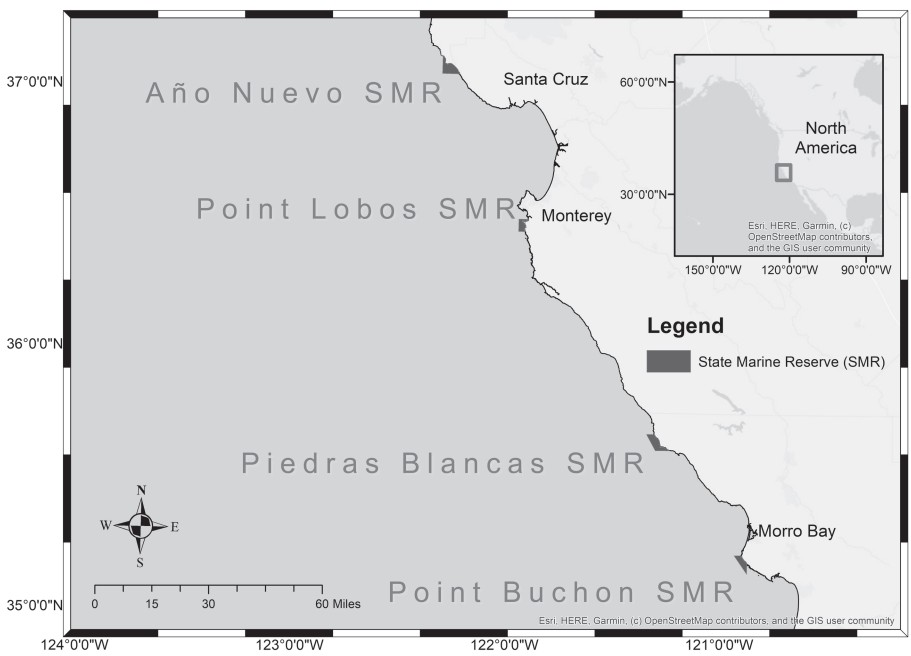

**Figure 1 Marine Protected Areas in central California monitored by CCFRP between 2007 and 2016.**
The base map shown, OpenStreetMap®, is *open data*, licensed under the Open Data Commons Open Database License and released under a CC-BY-SA license. Esri, HERE, Garmin © OpenStreetMap contributors.

The CCFRP program capitalizes on the expertise and knowledge of the fishing industry, angling public, participating scientists, and managers; together, these constituents work toward a common goal (e.g., measuring MPA effectiveness) that ultimately gives the group a shared purpose (*Wendt & Starr, 2009*; *Yochum, Starr & Wendt, 2011*). CCFRP relies on relationship building and transparency to create buy-in of MPA monitoring, evaluation, and management by involving stakeholders in all aspects of the program, from study design to data collection and sharing. An example of these efforts is the annual Volunteer Appreciation and Data Workshop hosted by the coordinating staff from California Polytechnic University, San Luis Obispo (Cal Poly) and Moss Landing Marine Labs (MLML) (*CA Collaborative Fisheries Research, 2018*), where survey results and trip highlights are shared with volunteer participants and other partners. To date, the impact of these events on angler opinions of MPAs has not been evaluated.

Human dimensions such as effective engagement, honesty, trust, and transparency can impact the success of MPAs (*Gall & Rodwell, 2016*; *Ordoñez-Gauger et al., 2018*). However, research on public knowledge, attitudes, and perceptions of California's MPA network is sparse, and studies that do exist vary widely across geographic region and composition of study populations (*Baldassare et al., 2007*, *2017*; *Loper, 2008*; *Ordoñez-Gauger et al., 2018*). While the success of CCFRP in generating valuable monitoring data is clear (*Wendt & Starr, 2009*; *Starr et al., 2015*), the degree to which CCFRP participation has influenced

volunteer perceptions of California's MPAs is less clear. In addition, the essential context for understanding this influence—(a) the demographics and characteristics of the CCFRP volunteer anglers and (b) their perceptions on the health of groundfish stocks, the data quality used to manage those stocks, and the effectiveness of MPAs relative to groundfish management measures (e.g., depth restrictions, bag limits, size limits)—has also not been assessed. Although California's network of MPAs were not specifically designed as a fishery management tool, any beneficial fisheries impacts of MPAs are important in the evaluation of overall MPA effectiveness. Thus, state resource managers would be well served by learning about the volunteer anglers who help monitor California's MPAs as well as their respective opinions on MPAs.

Given the longevity of CCFRP, and extensive volunteer participation, the program provides a valuable opportunity to measure outcomes of long-term stakeholder engagement. We used an online survey of current and former CCFRP volunteer anglers to learn about (a) volunteer angler demographics and attitudes toward groundfish management and stock health, (b) volunteer angler motivations for joining and staying in the program and (c) whether participation in the program influenced volunteer angler opinions on the quality of fisheries data used in resource management and the creation of MPAs in California. By characterizing the population of CCFRP angler volunteers and their perceptions in relation to their volunteer efforts, our intent is to characterize the realized benefits of CCFRP as a collaborative research program, beyond the fisheries data it yields.

## MATERIALS AND METHODS

### Survey

We distributed an online survey to 722 volunteer anglers who participated in CCFRP with Cal Poly and MLML between 2007 and 2018. This group represented a subgroup of the entire volunteer population during that time period ($N = 901$), as 179 volunteer anglers had previously opted out of receiving communications from CCFRP and two other groups, individuals without e-mail addresses and anglers under the age of 18 years, were not contacted. We used Qualtrics, an online survey platform, to deliver the survey questionnaire. Respondents provided written consent by agreeing to participate in the survey. The questionnaire consisted of 29 questions arranged into four sections: (a) CCFRP volunteering; (b) fisheries management and health of California groundfish stocks; (c) MPAs; and (d) demographics and miscellaneous questions (Article S1). We included multiple question types (yes/no, multiple-response, ordinal scale, and free-response) and designed the survey so that respondents could complete their responses in approximately 15 minutes. The University of California, San Diego Institutional Review Board (IRB) certified this study of volunteer anglers as exempt from IRB review.

We distributed the survey via a series of e-mails sent to subjects over a two-week period in Spring 2018. The first e-mail invited subjects to participate in the survey, and two subsequent e-mails sent seven and 12 days into the study period reminded subjects to complete the survey. Each e-mail contained a description of the study, a letter of consent, and a link to the online questionnaire.

## Acceptable survey response rate and margin of error

An acceptable survey response rate for categorical response surveys is dependent in part on the confidence level and the maximum margin of error that the surveyor is willing to accept (*Bartlett, Kotrlik & Higgins, 2001*). We calculated and report the acceptable minimum survey response rates according to the minimum sample sizes needed for an allowance of 5% and 10% margins of error (MOE$_{95}$) as described in *Bartlett, Kotrlik & Higgins (2001)*. Five and 10% MOE$_{95}$ are equivalent to ±0.25 and 0.5 points respectively, on a categorical ordinal response scale from 1 to 5. With respect to the survey response rate and estimates, the MOE$_{95}$ corresponds to the ± percentage points defining the range of the 95% *CI*, where,

$$\text{MOE}_{95} = z * \sqrt{(\hat{p} \times (1 - \hat{p}))/n}$$

$z$ = $z$-score for 95% *CI*,
$\hat{p}$ = sample proportion positive, and the second term in the equation is the standard error of a binomial distribution.

## Measures taken to address potential limitations with survey design

Potential limitations with the survey design included (1) the possibility a biased population of CCFRP volunteer anglers responded to the survey (i.e., nonresponse bias) (*Fisher, 1996*; *Bartlett, Kotrlik & Higgins, 2001*), (2) potential bias in respondent responses due to surveyor association with the CCFRP program (i.e., response bias) and (3) reliance on a respondent's ability to accurately recall the history and influence on their opinion change or lack thereof (e.g., response bias due to the subjectivity of a reflexive counterfactual study design) (*Franks et al., 2014*).

Steps taken to increase the survey response rate and aim for a large and representative sample included providing respondents (a) assurance of confidentiality, (b) a short, well-designed survey (e.g., ~15 min completion time) (c) a seamless online submission format, and (d) reminder emails (*Fisher, 1996*). Due to the anonymity of the survey we were unable to test (or adjust) for nonresponse bias. However, the survey questions provided a means to check whether respondents represented an unexpected demographic (e.g., mostly young anglers) as well as to compare the distribution of MPA opinion change responses by age, angler avidity, conservation-mindedness, level of engagement, etc.

With respect to potential surveyor influence on responses, the solicitation and reminder emails were sent via CCFRP field technicians (to keep volunteer emails confidential), but subjects were informed the survey itself was independently formulated by researchers at Scripps Institution of Oceanography, UC San Diego.

Questions related to opinion change and volunteer participation were included in separate sections of the survey so that these responses were made independent of each other. The longest time frame for which a respondent was asked to recall their opinions was dependent on the length of time since joining CFFRP, which at most, was 11 years (e.g., 2007–2017).

### Angler demographics and characteristics

Age and gender comprised the survey's demographic categories; other characteristics included years of fishing experience, frequency of fishing, degree of conservation-mindedness, whether anglers had any prior work experience in marine resource management or the recreational or commercial fishing industries, whether anglers had fished in MPA sites prior to those areas being designated MPAs, and whether anglers had ever participated in the MLPA planning process.

We categorized angling avidity (i.e., a relative measure of the enthusiasm an angler has for the sport) into three avidity levels—low, medium, or high—based on the number of saltwater angling trips they took per year, outside of CCFRP surveys, with low being <4 days, medium 4–23 days, and high >23 days per year. Angler avidity ranges were based on the National Oceanographic and Atmospheric Administration (NOAA) West Coast Fishing Avidity categories (*Rubio, Brinson & Wallmo, 2014*).

### Volunteer perceptions of groundfish management and stock health

In addition to characterizing the general opinion of anglers on the health of groundfish stocks and the effectiveness of specific regulations, we also compared the percentage of respondent opinions across related work experience categories to gauge the relative degree of consensus in opinions among these groups. Work experience categories included marine resource management, commercial or recreational fishing industry, and no experience.

### Volunteer opinion change on fisheries data quality and MPAs

Questions regarding the quality of fisheries data were limited to whether volunteers had an opinion of these data before CCFRP participation, and of those that did, whether their opinion changed either positively or negatively after having volunteered with CCFRP.

Given that MPAs were expected to elicit strong opinions with the angling public we were interested in (a) capturing the distribution of opinions on MPAs both before and after CCFRP participation, (b) characterizing the extent of opinion change across the group, and (c) examining whether opinion change is mediated by the extent of program participation. Respondents answered questions on an ordinal scale. They could report an opinion of "Positive", "Somewhat positive", "Somewhat negative", "Negative", or "No opinion". We coded the answers 1, 2, 3, 4 and 5 respectively.

To capture the overall proportion of respondents having a change in opinion on MPAs after volunteering with CCFRP, we subtracted the answer code corresponding to their opinion after CCFRP participation from the answer code corresponding to their opinion before volunteering with CCFRP. Results that were positive indicated a positive change in opinion of MPAs, results that were negative indicated a negative change in opinion of MPAs, and results that were "0" represented no change in opinion. We coded these differences numerically into a single variable representing change in opinion, with positive change coded as "1", no change coded as "2", and negative change coded as "3". We report the survey estimates $\pm MOE_{95}$ for the proportions of positive MPA opinions *before* and *after* CCFRP participation. Confidence intervals for which the calculated $MOE_{95}$ was

less than the minimum $MOE_{95}$ for our reported survey response rate were adjusted accordingly.

### Volunteer opinion change relative to measures of participation

To evaluate volunteer opinion change relative to levels of volunteer participation, we focused on three measures of CCFRP volunteer angler participation: (a) number of years since becoming a volunteer angler; (b) the number of CCFRP Volunteer Angler Appreciation and Data Workshops an angler had attended; and (c) approximate number of CCFRP sampling trips attended. We calculated the number of years since an angler became a volunteer by subtracting the year the respondent started volunteering from the year 2017 (the year prior to the on-line survey). Given the survey was anonymous, we calculated the approximate number of sampling trips a respondent went on throughout their time with CCFRP by multiplying the number of years a volunteer participated in CCFRP by the average number of trips they went on per year.

We used the glm function within the stats package in R version 6.3.1 (*R-Core-Team, 2019*) to run a binomial logistic regression model and test the effect of each measure of volunteer participation on respondents having either a positive opinion change or no opinion change on the quality of data used in resource management. There were too few ($n = 1$) negative opinion change responses to include this category within a multinomial logistic regression model (Data S1).

We used the nnet, broom, scales, and car packages (*Venables & Ripley, 2002*; *Fox & Weisberg, 2019*; *Wickham & Seidel, 2019*; *Robinson & Hayes, 2020*) in R version 6.3.1 (*R-Core-Team, 2019*) to run a multinomial logistic regression model and test the effect of each measure of volunteer participation on respondents having a positive, negative, or no opinion change on the creation of MPAs (Data S1). We used the MNLpred package (*Neumann, 2020*) to construct predicted probabilities and 95% confidence intervals for each opinion change category over levels of participation (Data S1).

Demographics and characteristics of respondents were also compared across MPA opinion change categories.

## RESULTS

Of the 722 current and former volunteer anglers contacted about the survey, 112 completed and submitted a survey (Data S1), for a response rate of 15%. One respondent had not yet volunteered on sampling trips, leaving 111 surveys included in the analysis. The acceptable minimum response rate, given a 10% $MOE_{95}$ was 12% ($n = 85$); for a 5% $MOE_{95}$, the minimum response rate was 35% ($n = 251$). Given our survey response rate, the minimum $MOE_{95}$ for reporting was 9%. Excluding one outlier (24.7 h), the average time respondents took to complete and submit the survey was 12.4 min (±6.4 SD).

### Volunteer demographics and characteristics

The distribution of respondent age was skewed left, with nearly one third of respondents being between 65 and 74 years of age, and the next largest age bracket being 55-64 years old (17%, Table 1). Most respondents were male (86%); twelve percent (12%) were female.

**Table 1 Demographics and characteristics of CCFRP volunteer angler survey respondents.**

| Category | Number of respondents | Percent of respondents |
|---|---|---|
| Age | | |
| 18–24 | 3 | 3% |
| 25–34 | 16 | 15% |
| 35–44 | 11 | 10% |
| 45–54 | 17 | 15% |
| 55–64 | 19 | 17% |
| 65–74 | 35 | 32% |
| 75+ | 9 | 8% |
| Gender | | |
| Male | 95 | 86% |
| Female | 13 | 12% |
| Prefer not to state | 2 | 2% |
| Angler avidity[a] | | |
| Low | 27 | 25% |
| Medium | 44 | 40% |
| High | 38 | 35% |
| Participated in the MLPA planning process | | |
| Yes | 20 | 18% |
| No | 90 | 82% |
| Conservation Mindedness[b] | | |
| More | 77 | 70% |
| Similar | 27 | 25% |
| Less | 1 | 1% |
| Related work experience[c] | | |
| Recreational fishing only | 12 | 11% |
| Commercial fishing only | 3 | 3% |
| Marine resource management only | 11 | 10% |
| Management and Commercial | 0 | – |
| Management and Recreational | 2 | 2% |
| Recreational and Commercial | 7 | 6% |
| All three | 2 | 2% |
| None | 71 | 65% |
| Fished and sampled at CCFRP sites before and after MPA creation | | |
| Yes | 36 | 33% |
| No | 74 | 67% |
| Total | 110 | 100% |

**Note:**
[a] The 1 respondent who did not answer this question was not included.
[b] The 5 respondents who did not answer this question were not included.
[c] The 2 respondents who had incomplete answers for these questions were not included.

Two percent (2%) of respondents chose the option "I prefer not to say." Respondents had medium to high avidity for saltwater angling, with 40% taking between four and 23 trips a year and 35% taking more than 23 trips a year. Eighteen percent (18%) of respondents said they participated in the MLPA planning process. Of those participating in the MLPA planning process, 90% were characterized as having high or medium angling avidity (45% each).

Seventy percent (70%) of volunteer anglers who responded to the survey considered themselves to be more conservation minded than their peers in the recreational fishing community, and an additional 25% thought they were similarly conservation minded with their peers (Table 1). Sixty-five percent (65%) of respondents did not have any experience in marine resource management, or the recreational or commercial fishing industries. A total of 24% had experience working in the fishing industry sector, and 14% had some experience with marine resource management (there was some overlap between these groups). Forty-five percent (45%) of the respondents had previously fished in areas that are now MPAs. Seventy-four percent (74%) of respondents participated in surveys of both MPA and reference areas during their time as volunteers with CCFRP.

## Volunteer perceptions of groundfish management and stock health

Many respondents (52%) believed California groundfish stocks were healthy, while 25% thought they were somewhat unhealthy or very unhealthy; the distribution of responses was similar across related work experience category (Fig. 2A). Nearly four out of five respondents (79%) thought that California groundfish stocks were very well managed, well managed, or adequately managed; 14% believed they were poorly managed, and 2% thought they were very poorly managed. The distribution of responses regarding the management of stocks varied by related work experience (Fig. 2B). Respondents having no related work experience and respondents with marine resource management experience had a more positive response toward the management of groundfish stocks than respondents having worked in the fishing industry (Fig. 2B).

Eighty-five percent (85%) of respondents thought seasonal closures and bag limits were effective fisheries management tools for groundfish stocks, while spatial closures and depth restrictions were considered relatively less effective (Fig. 3A). Negative responses toward the effectiveness of spatial closures and depth restrictions were comprised mostly of respondents with experience working in the fishing industry (Fig. 3B). Respondents with no experience and those with fishing industry experience were least certain about depth restrictions; this type of regulation comprised the highest percentage of "Not sure" responses in these groups (Fig. 3C).

## Motivations for volunteering with CCFRP

Nearly all respondents said they plan to continue volunteering with CCFRP (93%). Of those who plan to continue volunteering ($N = 102$), the reasons why they joined the program were the same as the reasons why they continue volunteering with the program (Fig. 4). The most frequently selected reason for continuing to volunteer with CCFRP was the opportunity to participate in science (75%). Sixty-eight percent (68%) selected

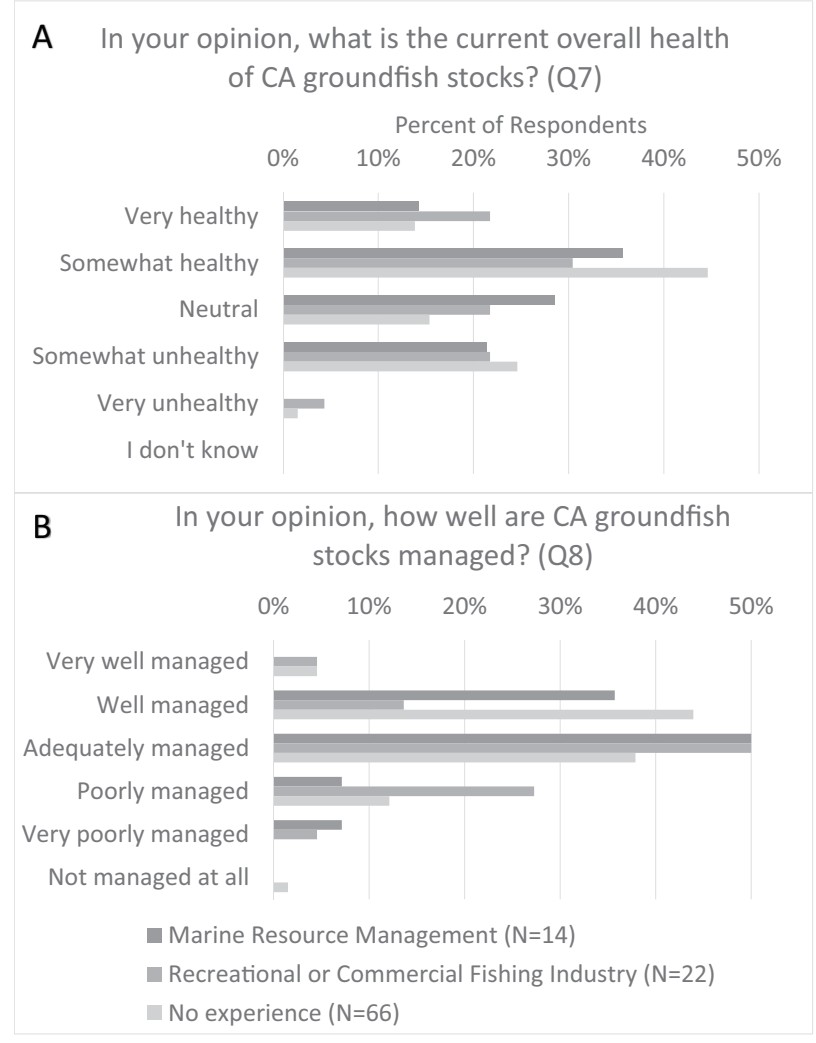

**Figure 2 Comparison of CCFRP volunteer angler opinions on California groundfish health and management relative to volunteer related work experience.** (A) Percentage distribution of survey respondent opinions on the health of California groundfish stocks. (B) Percentage distribution of survey respondent opinions on California groundfish management. The distribution of angler responses (*N* = 102) is reported relative to their related work experience (fisheries management, fishing industry, no experience).

"giving back to fisheries resources", and 58% selected "enjoying a day of fishing provided by CCFRP" as key reasons why they both joined the program and why they stay involved (Fig. 4). Several respondents who responded "Other" described desires to help fisheries or help marine resource managers gather data to use in management. Three respondents replied that the opportunity to learn was important to why they joined the program, and an additional three respondents cited learning new information as a reason for continuing to volunteer.

Eight respondents (7%) said that they do not plan to continue volunteering with CCFRP. Reasons included lack of available volunteer spots (*n* = 2), personal health (*n* = 2),
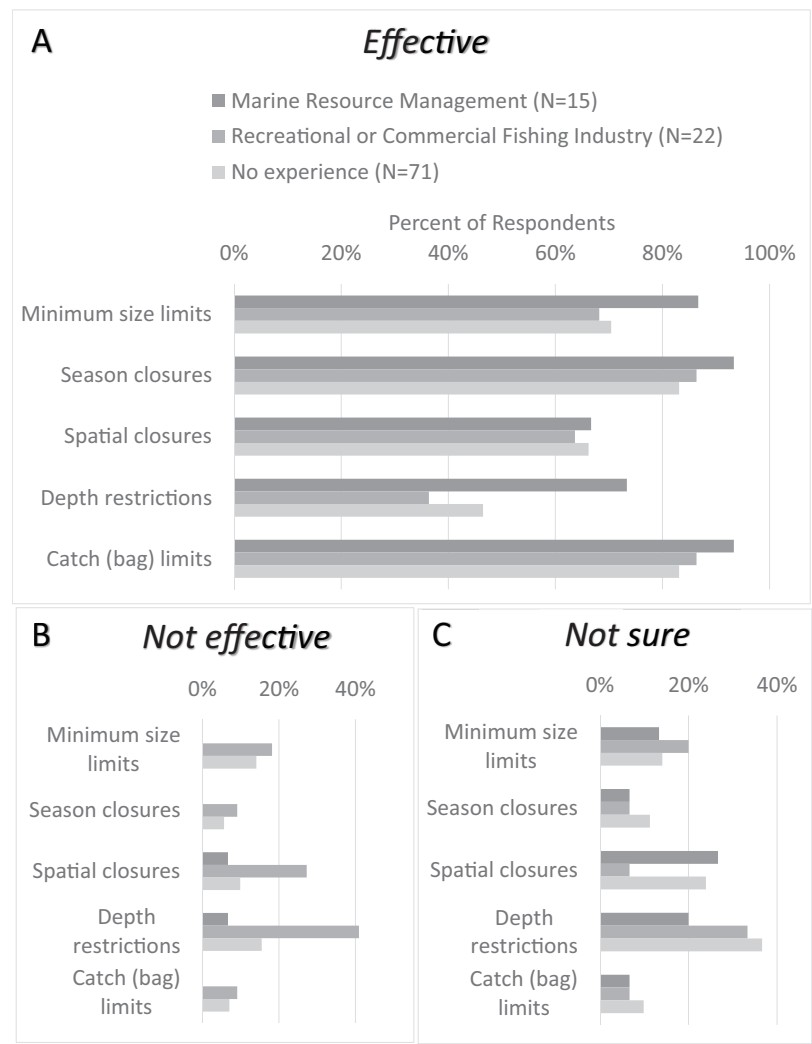

**Figure 3 Comparison of CCFRP volunteer angler opinions on California groundfish management strategies relative to their related work experience.** (A) Percentage distribution of survey respondents who believe California groundfish management strategies to be "Effective". (B) Percentage distribution of those who believe them to be "Not effective". (C) Percentage distribution of those who responded "Not sure." The distribution of angler responses ($N = 108$) is reported relative to their related work experience (fisheries management, fishing industry, no experience).

seasickness ($n = 1$), old age ($n = 1$), turned into a job ($n = 1$), and a lack of extra time to volunteer ($n = 1$).

## Opinion change on fisheries data quality and MPAs

Sixty-one percent (61%) reported having no opinion of the quality of fisheries data used for resource management before volunteering with CCFRP; roughly equal portions reported having either no change (18%) or a positive change in opinion (20%) after volunteering with CCFRP; 1% reported a negative change in opinion (Fig. 5).

Sixty percent (60%) of volunteer anglers surveyed said that they had positive or somewhat positive opinions of the creation of MPAs before they began volunteering
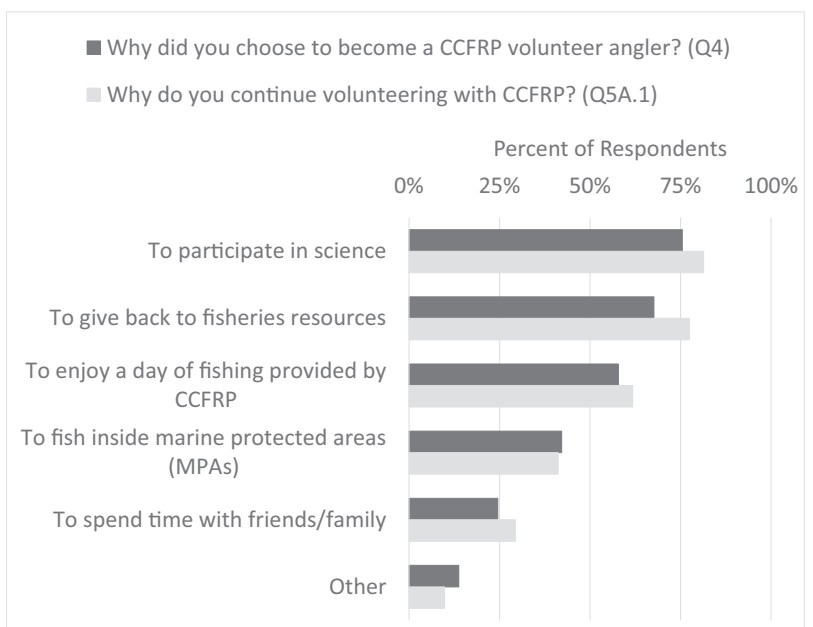

**Figure 4 Volunteer angler motivations to join CCFRP and to continue volunteering with the program.** Percentage distribution of the reasons why survey respondents joined CCFRP and why they continue with the program (only showing responses of volunteers who plan to continue, $N = 102$).

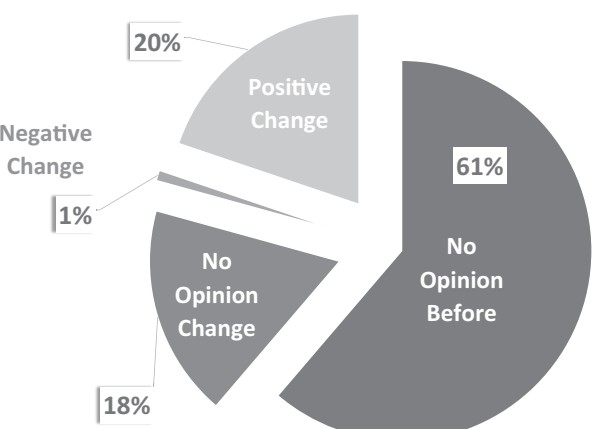

**Figure 5 CCFRP volunteer angler opinion change on the quality of fisheries data.** Percent of CCFRP volunteer angler survey respondents ($N = 110$) who had no opinion on the quality of fisheries data used in resource management before becoming CCFRP volunteers, no change in opinion after volunteering with CCFRP, and either a positive or negative change in opinion after volunteering with CCFRP.

(Fig. 6A). The $MOE_{95}$ for these responses was $\pm 9\%$ ($CI_{95} = 51–69\%$), which was within the $MOE_{95}$ for our survey. Twenty-eight percent (28%) said they had somewhat negative or negative opinions of MPA creation in California before volunteering, while 15% of respondents said they did not have any opinion of MPAs before joining the program
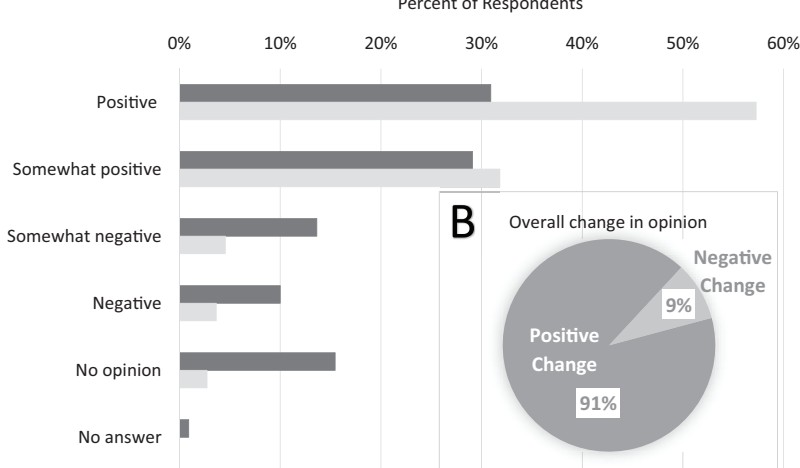

Figure 6 **CCFRP volunteer angler opinions on MPAs.** (A) Percentage distribution of survey respondent opinions on MPAs before and after volunteering with CCFRP, $N = 110$. (B) Overall percent of survey respondents having a positive or negative change in opinion on MPAs after becoming CCFRP volunteers.

(Fig. 6A). When volunteers were asked what their opinions were after volunteering with CCFRP, 89% said they had a positive or somewhat positive opinion of MPAs (Fig. 6A). The $MOE_{95}$ for these responses was ±6% ($CI_{95}$ = 83–95%); however, after adjusting for the minimum $MOE_{95}$ (±9%) for our survey response rate, the $CI_{95}$ becomes 80–98%. The proportion of respondents having no change of opinion on the creation of MPAs after volunteering with CCFRP was 49%; these respondents comprised 95% of those having a positive or somewhat positive opinion before participating with CCFRP. Of those respondents having a change of opinion ($n = 57$), 91% had a positive change and 9% of respondents had a negative change in opinion of MPAs after volunteering with CCFRP (Fig. 6B).

## Measures of volunteer participation

The number of years volunteers participated with CCFRP was nearly uniformly distributed; volunteers who had been with the program since 2007 made up the highest percentage (15%) and newly recruited volunteers (in 2017) followed behind at 12%. Fifty-four percent (54%) of volunteers surveyed never attended an annual Volunteer Appreciation and Data Workshop. Of the 46% who had, most attended one to four workshops. Six percent (6%) of respondents attended five or more workshops. The estimated number of CCFRP trips attended ranged from one trip to 154 trips (median = 8 trips, mean = 17 trips). Seventeen percent (17%) of respondents attended one sampling trip.
**Table 2 Likelihood ratio tests of the binomial logistic regression model of change in opinion on the data quality used in resource management relative to measures of volunteer angler participation with CCFRP.**

| Predictors | Likelihood ratio tests | | |
| --- | --- | --- | --- |
| | Chi-Square | df | *p* |
| Length of time since joining CCFRP | 0.320 | 1 | 0.572 |
| Number of Volunteer Appreciation and Data Workshops attended | 0.713 | 1 | 0.340 |
| Number of sampling trips with CCFRP | 0.000 | 1 | 0.977 |

**Table 3 Likelihood ratio tests of the multinomial logistic regression model of change in opinion on MPAs relative to measures of volunteer angler participation with CCFRP.**

| Predictors | Likelihood ratio tests | | |
| --- | --- | --- | --- |
| | Chi-square | df | *p* |
| Length of time since joining CCFRP | 12.8658 | 2 | *0.002* |
| Number of Volunteer Appreciation and Data Workshops attended | 0.951 | 2 | 0.622 |
| Number of sampling trips with CCFRP | 0.124 | 2 | 0.940 |

**Note:**
Italics denote significant *p*-values.

## Volunteer opinion change relative to measures of participation

The analysis on the quality of fisheries data used in resource management was limited to only those anglers who stated they had an opinion before volunteering with CCFRP ($n = 42$). This is because volunteers who did not have an opinion prior to volunteering were not asked about their opinions after volunteering with CCFRP (see Methods). None of the measures of volunteer participation were significantly related to having a positive change in opinion (versus no change) on data quality (Table 2).

One hundred and seven (107) of the respondents answered all questions related to MPA opinion change and the three calculated measures of participation (length of time since joining the program, number of Volunteer Appreciation and Data Workshops attended, and total number of trips attended). Length of time since joining CCFRP was the only significant predictor of having a change in opinion regarding MPAs (Table 3). In general, as the time since joining CCFRP increased, a volunteer angler was more likely to have a positive change in opinion on MPAs than having no change in opinion ($RRR = 0.82$ (reference category = positive change in opinion), 95% CI [0.72–0.92], $z = −0.293$, $p = 0.003$; Fig. 7).

## MPA opinion change by volunteer characteristics

The distribution of respondents within different volunteer characteristics, including angler avidity, conservation mindedness, and related work experience, were similar across MPA opinion change categories (Table 4); however, respondents who expressed no opinion change of MPAs tended to be younger than those who had a positive change in opinion of MPAs. Those respondents who had previously worked in marine resource management

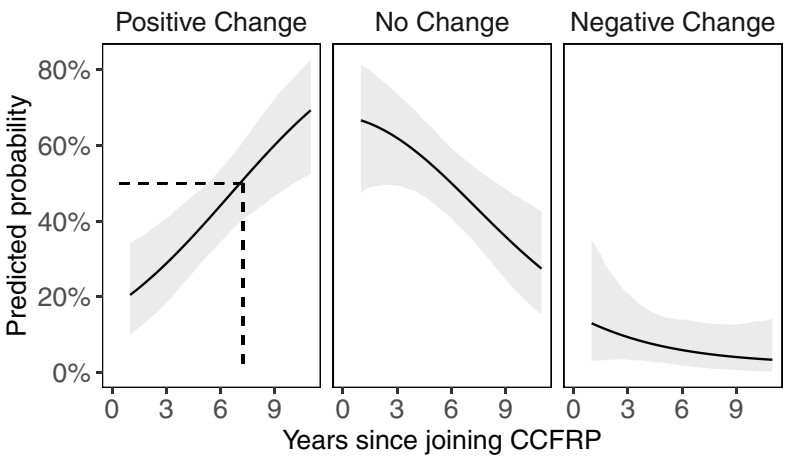

**Figure 7 Predicted probability of CCFRP volunteer anglers having an opinion change on MPAs relative to time.** Predicted probability (±95% CIs) of a CCFRP volunteer angler having a positive change in opinion, no change in opinion and negative change in opinion on MPAs relative to the length of time (years) since joining CCFRP. The dashed line relates the length of time when the probability of having a positive change in opinion on MPAs becomes greater than or equal to 50% (=7.25 years).

were split between having no change in (positive) opinion and having a positive change in opinion of MPAs.

The sample size for those who expressed a negative change of opinion toward MPAs consisted of five respondents (Table 4). Of these, none participated in the MLPA planning process or worked previously in marine resource management. Three had fished in both MPAs and reference sites with CCFRP, but none had visited the same MPA sites with CCFRP that they had fished in before the implementation of MPAs in 2007.

## DISCUSSION

Our results provide evidence that long-term engagement of stakeholders in collaborative research can positively change angler opinions on MPAs. At the outset, the creators of CCFRP postulated that collaborative research was a "potent mechanism" that could (among other listed benefits) build trust in fisheries management and develop a more accurate consensus about resource status (*Wendt & Starr, 2009*). These anticipated outcomes are directly linked to the collaborative nature of the program, where participants are working together toward a shared goal (*Wendt & Starr, 2009*; *Yochum, Starr & Wendt, 2011*). CCFRP straddles two modes of public engagement in science: collaborative fisheries research and citizen science. In so doing, it draws from a long history of scientists partnering with members of the fishing industry to study fish populations or develop management tools (*Hartley & Robertson, 2009*; *Mireles, Nakamura & Wendt, 2012*; *Gleason, Iudicello & Caselle, 2017*). Citizen science – also called community-based or participatory science – involves members of the public who are not scientists by trade (*Mckinley et al., 2017*), and it differs from collaborative fisheries research in that the volunteers are not necessarily part of the fishing industry. While the partnership between

**Table 4 Distribution of CCFRP volunteer angler survey respondents by angler demographic within each MPA opinion change category.**

| Category | Positive change (N = 51) | No change (N = 53) | Negative change (N = 5) |
|---|---|---|---|
| **Age** | | | |
| 18–24 | – | 6% | – |
| 25–34 | 14% | 15% | 20% |
| 35–44 | 6% | 11% | 20% |
| 45–54 | 12% | 21% | – |
| 55–64 | 18% | 19% | – |
| 65–74 | 41% | 23% | 40% |
| 75+ | 10% | 6% | 20% |
| **Angler avidity**[a] | | | |
| Low | 24% | 26% | 20% |
| Medium | 41% | 36% | 60% |
| High | 35% | 38% | 20% |
| **Conservation Mindedness**[b] | | | |
| More | 63% | 77% | 60% |
| Similar | 29% | 21% | 20% |
| Less | 2% | – | – |
| **Related work experience**[c] | | | |
| Recreational fishing only | 6% | 15% | 20% |
| Commercial fishing only | 6% | – | – |
| Recreational and commercial | 12% | 2% | – |
| Marine resource management only | 14% | 8% | – |
| Management and recreational | – | 4% | – |
| All three | – | 4% | – |
| None | 63% | 66% | 60% |

**Notes:**
[a] The one respondent who did not answer this question was not included.
[b] The five respondents who did not answer this question were not included.
[c] The two respondents who had incomplete answers for these questions were not included.

CCFRP and CPFVs follows a more traditional collaborative fisheries research model, the inclusion of the angling public distinguishes CCFRP as having successfully integrated citizen science into collaborative fisheries research.

## CCFRP volunteers are mostly older, avid anglers

California Collaborative Fisheries Research Program volunteers who responded to our survey were representative of fresh and saltwater anglers in California (mostly men); however, they were relatively older. Forty-nine percent (49%) of the larger angling community are between 18 and 44 years old and less than 4% are 65 or older (*U.S. Department of the Interior et al., 2011*). In contrast, 18–44-year-olds made up less than a third of our angler respondents, and 40% were over the age of 65. CCFRP surveys occur only on weekdays, of which older, retired adults are more likely to be free for volunteering compared to younger anglers. This older demographic may have influenced the

proportional distributions of certain volunteer characteristics such as angler avidity (i.e., more time for fishing opportunities) and perceptions that could be influenced by having a more historical perspective (e.g., stock health). Our survey did not include questions regarding household income or ethnicity.

Relative to saltwater recreational anglers on the West Coast of the United States (*Rubio, Brinson & Wallmo, 2014*), CCFRP volunteer anglers surveyed in our study had higher fishing avidity, having on average participated in a higher number of fishing trips (non CCFRP-related) in the last year. Anglers with high fishing avidity have a greater stake in fisheries management decisions. For instance, in a 2014 survey of saltwater recreational anglers, angler avidity was positively correlated with perceived importance of ensuring "that the opinions of all recreational fisheries stakeholders are considered in policy-making" (*Rubio, Brinson & Wallmo, 2014*). While our volunteers were not asked to report their opinions on the importance of stakeholder input in policy making, we found that avid CCFRP volunteer angler respondents were more likely to have participated in the MLPA planning process.

Levels of public participation in the California MLPA planning process were very high, with over 4,000 members of the public attending planning-related events and over 70,000 public comments submitted during the process and environmental review (*Gleason et al., 2013*). Still, with over 39 million residents in California (*United States Census Bureau, 2017*), this is a relatively small proportion of participants. In our study, one in five CCFRP survey respondents participated in the MLPA in some form, making them more engaged than the average resident. Perhaps not surprisingly, about one third of our respondents who participated in the MLPA were marine resource managers, however not all of those who had worked in marine resource management participated in the MLPA.

We found that CCFRP has successfully engaged members of the public, as two thirds of respondents had no work experience related to either marine management or the fishing industry. Although the general audience targeted for volunteer angler recruitment was recreational anglers (*Wendt & Starr, 2009*), the experience of fishing side-by-side with people from different professional backgrounds may aid in the relationship-building that is an important cornerstone of the program.

Low survey response rates can introduce nonresponse bias in survey results if the respondents are not characteristic of the overall survey population (*Fisher, 1996*; *Bartlett, Kotrlik & Higgins, 2001*). Due to the relatively low survey response rate in this study (15%), non-response bias is not precluded from our results (*Fisher, 1996*). Nevertheless, the dominant characteristics of respondents (e.g., older men with high fishing avidity and no related work experience), are not atypical of the general CCFRP volunteer population. Many volunteer anglers reported being more conservation-minded than their peers in the recreational fishing community. This characterization is also not entirely unexpected given that CCFRP volunteer anglers are citizen scientists participating in collaborative fisheries research, of which conservation is a common motivator.
## CCFRP volunteer anglers are motivated by science and conservation

Part of CCFRP's success in relationship building is evidenced by the willingness of anglers to want to continue to participate in the program year after year. Most respondents said they plan to continue volunteering. The reasons respondents chose to stay with the program were the same three reasons they cited for joining CCFRP in the first place: (a) to participate in science; (b) to give back to fisheries resources; and (c) to enjoy a day of fishing provided by CCFRP. These responses demonstrate that CCFRP anglers are not solely driven by the novelty of fishing inside MPAs, but by their interest in being involved in fisheries research. A handful of respondents were high school teachers who responded that learning was a motivator for why they joined. Three other respondents listed learning as a motivator for why they stayed. Across marine and coastal citizen science projects, increasing knowledge is often a frequent motivation for volunteering (*Thiel et al., 2014*).

## Volunteer angler consensus on groundfish health and management

A lack of transparency between fishery managers and the fishing community has often led to angler distrust of fishery assessments and management measures (*Yochum, Starr & Wendt, 2011*); thus, one goal of collaborative fisheries research is garnering accurate consensus among the fishing community and fisheries scientists regarding resource health (i.e., everyone's perception of stock health reflects reality). In this study, most CCFRP respondents, regardless of their related work experience (including no related experience), believed groundfish stocks were somewhat healthy. This is a relatively accurate assessment considering most species of fish comprising the groundfish fishery in California are rockfishes, of which many stocks have rebuilt or are rebuilding from an overfished status (*NOAA Fisheries, 2019*; *Pacific Fishery Management Council, 2008*). The agreement that groundfish stocks are somewhat healthy, regardless of related work experience, suggests that there is accurate consensus of resource status among these groups. Although not explicitly addressed in our survey, it seems likely that CCFRP volunteer participation influenced these angler perceptions over time. It is also possible CCFRP volunteer perceptions regarding groundfish stock health are influenced by historical perspectives, as older respondents are more likely to have participated in groundfish fishing prior to the collapse and subsequent recovery of many rockfish stocks. In another survey of the iconic saltwater bass fishery in southern California, fishermen with more years of experience (and typically older in age) were more likely to have an accurate perception of stock health (*Bellquist et al., 2017*). In our survey, the proportion of younger respondents who had a "neutral" opinion regarding groundfish stocks was higher than that of the older respondents.

Most (79%) respondents thought groundfish stocks were well-managed. Many (65%) believed spatial closures (including MPAs) were effective in ensuring healthy groundfish stocks in California, though catch limits and season closures had higher support (85% each); 21% were "unsure" and 14% believed spatial management to be "not effective". Most of the uncertainty and negative opinion of spatial management was by respondents having worked in the fishing industry. However, depth restrictions were least popular among all related work experience categories and garnered the greatest amount of

uncertainty. Depth restrictions for groundfish in central California prohibit fishing in waters greater than 50 fathoms (91.4 m) and were intended to assist in rebuilding overfished rockfish stocks such as Canary Rockfish (*Sebastes pinniger*) and Yelloweye Rockfish (*Sebastes ruberrimus*). However, fishing these depths for other popular recreational groundfishes in central California (e.g., Lingcod (*Ophiodon elongatus*), Cabezon (*Scorpaenichthys marmoratus*), and Greenlings (Family Hexagrammidae)) is also precluded by this regulation, and could be driving some of the uncertainty among respondents. In addition, although Canary Rockfish was rebuilt in 2015 (*Thorson & Wetzel, 2015*), Yelloweye Rockfish remains in rebuilding status (*Gertseva & Cope, 2017*). Interestingly, except for depth restrictions, the relative proportion of respondents stating groundfish management measures are effective was similar across regulations and related work experience categories.

The focus of CCFRP is not to educate anglers on groundfish management and regulations. However, because groundfish regulations include mandatory release of overfished rockfish species, CCFRP does actively work to increase angler awareness of the susceptibility of rockfishes to pressure-related (i.e., depth-related) injuries associated with angling and the utility of recompression (i.e., releasing fish back to depth). Generally, fishing deeper results in an increased susceptibility to barotrauma and decreased survival rates of rockfishes; thus, CCFRP protocol has always restricted captains to fish areas in depths less than 36.7 m (120 ft). Additionally, CCFRP science crew release fish showing signs of barotrauma back to depth with descending devices since recompression alleviates signs of barotrauma and significantly increases release survival of many rockfishes (*Jarvis & Lowe, 2008*; *Hannah, Rankin & Blume, 2012*). These measures ultimately promote ethical rockfish angling practices.

## Volunteers are less opiniated on fisheries data quality than MPAs

In addition to outreach, a typical day on the water provides CCFRP volunteers opportunities to observe how data are collected. Important survey protocol details are relayed to CCFRP volunteer anglers on each day's pre-survey briefing. At the end of the day, the science crew debriefs the anglers on overall fish count, fish counts by angler, and biggest and smallest fish caught, etc. Thus, although the anglers do not assist with recording data, the anglers are immediately able to informally verify the data collected that day, based on their own observations and recollections.

Unlike the topic of MPAs, most respondents (61%) stated they did not have an opinion of the fisheries data used in resource management prior to volunteering for CCFRP. After participation with CCFRP, opinion change was mostly positive, but it remains unclear the degree to which this has to do with CCFRP. Although none of the metrics of angler participation were significantly related to positive change (versus no change) in opinion of fisheries data quality, our analysis was limited by a reduced sample size because (unlike the MPA analysis) only anglers who stated they had an opinion before volunteering with CCFRP were asked about their opinion change. Thus, we do not know whether anglers who had no opinion on data quality before volunteering with CCFRP eventually gained a positive or negative opinion, or what the opinions were of those not having an

opinion change. Nevertheless, the mostly positive opinion change suggests CCFRP participation may be a factor, regardless of the level of engagement. Building trust in the quality of fisheries data used for management is an important step toward increasing angler perceptions of groundfish management measures, including MPAs. It is also worth noting that anglers with high avidity serve CCFRP by providing highly experienced angling services, and likely relatively high consistency in angler skill levels, all positively influencing data quality.

## CCFRP positively influences opinions on MPAs

A significantly higher percentage of volunteer anglers surveyed had positive opinions of the creation of MPAs after volunteering with CCFRP. We did not find that this response was biased with respect to angler characteristics; the distribution of anglers across categories of angler avidity, conservation-mindedness, and related work-experience were similar, regardless of the direction of MPA opinion change. For example, although the majority of CCFRP volunteers responding to our survey identified themselves as being more conservation-minded than their peers in the recreational fishing community, about half of them gained a positive opinion of the creation of MPAs after volunteering with CCFRP. Thus, even those considering themselves to be conservation-minded did not necessarily have strong positive opinions of MPAs before participating with CCFRP. We also found that respondents varied in their level of engagement with CCFRP across all three different measures of participation; thus, survey respondents are not likely to be more engaged than the overall population of CCFRP volunteers. In fact, the wide range of engagement among respondents allowed us to test how different levels of participation related or not to MPA opinion change.

Respondent subjectivity can be a disadvantage of reflexive counterfactual survey designs (i.e., before and after opinions), in which there is no true control group and there is reliance on respondents to recall changes in their beliefs and opinions (*Smallhorn-West et al., 2019*, *Franks et al., 2014*). For example, we have no measure of how respondent opinions on MPAs would have changed had volunteers never participated in CCFRP, nor do we have an indication of the accuracy or legitimacy of subjective volunteer responses. However, the strength of our study design is that it allowed us to test what aspect of participation and the extent to which that volunteer participation was a factor in volunteer-stated opinion change. In other words, we did not solely rely on respondent before and after opinions to evaluate CCFRP influence on MPA opinions. In addition, we were interested in volunteer angler beliefs despite the potential for subjectivity. Capturing volunteer beliefs and perceptions is also important for highlighting opportunities for additional outreach and education (*Franks et al., 2014*).

The increase in positive perceptions of MPAs of CCFRP volunteers mirrors the perceptions of California's public. In 2017, more than three in four Californians said that it was very important that California have MPAs; a 20 point increase since 2006 (*Baldassare et al., 2007*, *2017*). While the overall increase in support for MPAs across the state in the last ten years might be considered a counterfactual outcome suggesting no effect of CCFRP on volunteer opinions of MPAs (*Smallhorn-West et al., 2019*), our study

results indicate the time spent volunteering for CCFRP was influential in volunteer opinion change.

## Time with CCFRP influences positive change of opinion on MPAs

A positive change of opinion toward MPAs was directly related to the number of years since respondents joined CCFRP. Other measures of participation, including the number of Volunteer Appreciation and Data Workshops or the number of CCFRP trips attended, were not significantly related to MPA opinion change, indicating that change in angler perceptions takes time. In this study, the length of time necessary to achieve a greater than fifty percent (50%) probability of having a positive change in opinion on MPAs was about seven years since joining CCFRP. Long-term stakeholder engagement with CCFRP corresponds with a longer period directly and indirectly gaining knowledge and awareness of MPAs through participation in survey trips and through CCFRP communications, including e-mails, e-newsletters, and posts on social media. Although Volunteer Appreciation and Data Workshops are arguably an important part of CCFRP's relationship building and outreach tools, it is often lived experiences that are more salient and have more impact on people's knowledge, attitudes, and perceptions.

Although not a stated goal of the study, we tested a posteriori whether any of the different measures of volunteer participation were perhaps related to a volunteer's willingness to continue participating (or not) with CCFRP (e.g., were volunteers who participated in more trips more likely to state they would continue volunteering?). While volunteers were significantly more likely to state they would continue volunteering with CCFRP than not continue (~13x more likely), none of the measures of participation were significant predictors of their willingness to continue with the program (Data S1). This would suggest that even newly recruited and less engaged volunteer anglers are enthusiastic in their support of CCFRP.

In 2017, CCFRP was expanded statewide, and now includes a partnership of six academic institutions that lead and organize surveys to actively monitor 14 MPAs in California (*Moss Landing Marine Laboratories, 2020*). Between 2017 and 2019, eight-hundred and ninety-eight (898) CCFRP volunteer anglers assisted science crew and CPFV captains/crew in surveying 77,202 fish representing 94 species statewide (R Brooks, 2020, personal communication). This large expansion of the program offers additional opportunity to learn about (a) demographics and characteristics of the fishing industry sector of CCFRP (CFPV captains and crew), (b) how demographics and characteristics compare by region within and among stakeholder groups, and (c) whether CCFRP has had differential influence on MPA perceptions across stakeholder groups. Bringing increased awareness of the human dimensions of stakeholders involved in collaborative fisheries research can only serve to continue to build relationships, create buy-in on management measures, and offer insights into areas of outreach that may need improvement.

## CONCLUSIONS

Our survey highlights CCFRP as a model for incorporating citizen science into collaborative fisheries research by capturing the realized benefits of collaborating with the

angling public. We have a clearer view of who CCFRP volunteers are as a group, and how participation in the program has shaped their perspectives. CCFRP volunteers are older and have a higher fishing avidity than the broader recreational angling community in California. Although they represent a heterogeneous group in terms of experience with related industry sectors, their perceptions of groundfish stock health and management are generally in agreement. Overall, these volunteers have a positive view of the fisheries data collected for resource management and the MPAs they help to monitor. This can be attributed, in part, to long-term participation in the program. Most notably, a positive change in opinion on MPAs was more likely to occur only after considerable time engaged with CCFRP (i.e., 7+ years). Future endeavors to develop new citizen science partnerships with collaborative fisheries research programs, in which to achieve similar benefits as CCFRP (e.g., building stewardship and advocacy), should focus not only on recruiting as many volunteers as possible, but in retaining those volunteers for as long as possible.

## ACKNOWLEDGEMENTS

This manuscript is based on A. Kellum's Master of Advanced Studies (MAS) Capstone Project at Scripps Institution of Oceanography, UC San Diego. E. Brazier assisted with survey design and analysis. T. Courtney, L. Waterhouse, N. Rosen, A. Siddall, L. Bellquist and L. Oremland provided technical expertise and advice. The staff, captains, and crews of Central Coast Sportfishing, Morro Bay Landing, Patriot Sportfishing, and Virg's Landing provided research survey support to Cal Poly while those of Tigerfish Sportfishing, Huli Cat Sportfishing, Kahuna Sportfishing, and J & M Sportfishing provided survey support to MLML. P. Smallhorn-West and one anonymous reviewer provided comments that greatly enhanced the manuscript. We are indebted to all the Central Coast CCFRP volunteer anglers, past and present, for their dedication to CCFRP and to those respondents who took the time to participate in our survey.

### Funding

This work was supported by a NOAA Fisheries Quantitative Ecology and Socioeconomics Training (QUEST) Program grant to Brice Semmens. There was no additional external funding received for this study. The funders had no role in study design, data collection and analysis, decision to publish, or preparation of the manuscript.

### Grant Disclosures

The following grant information was disclosed by the authors:
NOAA Fisheries Quantitative Ecology and Socioeconomics Training (QUEST) Program.

### Competing Interests

The authors declare that they have no competing interests.

## Author Contributions

- Erica T. Mason conceived and designed the experiments, analyzed the data, prepared figures and/or tables, authored or reviewed drafts of the paper, and approved the final draft.
- Allison N. Kellum conceived and designed the experiments, performed the experiments, analyzed the data, prepared figures and/or tables, authored or reviewed drafts of the paper, and approved the final draft.
- Jennifer A. Chiu conceived and designed the experiments, authored or reviewed drafts of the paper, and approved the final draft.
- Grant T. Waltz conceived and designed the experiments, authored or reviewed drafts of the paper, and approved the final draft.
- Samantha Murray conceived and designed the experiments, authored or reviewed drafts of the paper, and approved the final draft.
- Dean E. Wendt conceived and designed the experiments, authored or reviewed drafts of the paper, and approved the final draft.
- Richard M. Starr conceived and designed the experiments, authored or reviewed drafts of the paper, and approved the final draft.
- Brice X. Semmens conceived and designed the experiments, authored or reviewed drafts of the paper, and approved the final draft.

## Human Ethics

The following information was supplied relating to ethical approvals (i.e., approving body and any reference numbers):

The University of California, San Diego Institutional Review Board (IRB) certified this study of volunteer anglers as exempt from IRB review.

## Data Availability

Survey data and code are available online in a GitHub repository: https://github.com/ETJarvisMason/mpa-opinions. Data and code are available as a Supplemental File.

## Supplemental Information

Supplemental information for this article can be found online at http://dx.doi.org/10.7717/peerj.10146#supplemental-information.

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
