# Peer review of "Long-term participation in collaborative fisheries research improves angler opinions on marine protected areas"

_PeerJ, doi:10.7717/peerj.10146_

## Round 0.1 · original submission · Major Revisions

This is a well written article and provides useful insight to an interesting topic. Overall both reviewers have provided positive comments, but have also provided some feedback that would best if it was addressed and then resubmitted for further review.

Please pay particular attention to the following reviewer comments:
Reviewer 1 commented on your conclusions about whether long term participation improves opinions on fisheries data quality (51% changed opinion). While some moderation of language may be necessary, I wonder if a bar plot could be incorporated to support Fig 5, similar to the bar plot incorporated into Fig 6 (the results of which this reviewer found more convincing).

Both reviewers suggested the need for a greater mention of caveats/assumptions/shortcomings of the data generated by your study approach. For example, is it possible that more conservation-minded volunteered were more likely to respond to the survey, the study relies on respondents understanding causal links to their change in opinions and having memory of their opinions too? As suggested by Reviewer 1, these suggestions might present an opportunity to incorporate a section in the discussion focusing on the limitations of the study. This might also allow for a broader incorporation of the literature to support this section, and also to develop the Discussion overall.

·

Basic reporting

See below

Experimental design

See below

Validity of the findings

See below

Additional comments

This manuscript looks at the influence of participation in the CCFRP on the perceptions of anglers towards MPA management and fisheries data. It uses an online survey of 111 respondents to examine their self-reported changes in opinions. Overall the manuscript is very well written and structured and I think would make a valuable contribution to the field. It touches on an important point, the fact that citizen science isn’t just about the science but also public outreach, and sets out to test hypothesis in line with this. I think the conclusions about the importance of long-term involvement are really good to share.

There are however a few major issues that I think need to be resolved before this manuscript could be accepted. Therefore I am recommending a major revision. Below are a series of points that require addressing. While most are relatively minor, there are two issues that are major.

Major comments:
1. First and foremost, I don’t think your data supports your conclusions about whether long term participation improves opinions on fisheries data quality. Looking at figure 5, it’s hard to really accept your conclusions when only 51% of those with preexisting opinions changed towards the positive. The nature, and limitations, of survey data like this is that it is difficult to analyze (i.e. there are no means and SD here that you can test for this specific question), therefore I don’t think it’s really fair to say that the program improved opinions on fisheries data quality. Given there were 111 participants, and how close these numbers are, there is no way to be certain that if more people were surveyed, or less, that the 51 vs 49% wouldn’t change. I therefore think the entire manuscript, from the title to the discussion, needs to be adjusted accordingly. Figure 6, and the opinions about MPA looks solid, but not the results presented in figure 5. And as your title claims, this is effectively half of the story.

2. The second major point has to do with limitations from these kinds of studies, which I don’t think was adequately addressed anywhere in the manuscript. I would like to see a section, perhaps in the discussion, mentioning the caveats or limitations of this approach and data. The issue has to do with attributing causation that changes in opinions occurred as a direct result of the program, and not other confounding factors. This study uses a reflexive counterfactual (See reference below for definition, although there are other papers which may be more suitable to reference), whereby the study relies on each respondent having a clear understanding of causal links. i.e. the issue here is that after the fact, asking people whether a program has been the thing which has changed their opinion relies on them understanding other potential factors that may have also changed their opinion, and also accounting for whether they were already conservation minded. In addition to the fact that the program itself is the one doing the asking, which may bias peoples answers. In a perfect world your hypothesis would have been tested by sampling behaviors (or opinions of management, but not opinions about changes in opinions to management) prior to and following their participation in the program to look for differences. In this case this wasn’t possible, which is okay, but I think this needs to be addressed as a limitation of this kind of data. It is relying on peoples sometimes imperfect memories, after the fact and asked by a group with a potential conflict of interest. Within this, a second caveat is the fact that a biased group of people likely answered the survey, they had to be engaged enough to answer it, so it could have easily been a biased population. I don’t think all of this warrants disregarding the results, but these limitations should be articulated, and the conclusions adjusted for these caveats.

Smallhorn-West P, Weeks R, Gurney G, Pressey B (2019) Ecological and socioeconomic impacts of marine protected areas in the South Pacific: assessing the evidence base. Biodiversity and conservation


Minor comments:
• Reference for figures should be at the start of when they are mentioned, not at the end (throughout, but e.g. fig 2b should be on line 233, not 236)
• Figures 2 and 3: I think you could change the shading order. If darkness is meant to be hypothesized conservation knowledge, then dark would be marine resource management, mid fishing industry and light no experience or vice versa.
• Figure 3: If possible put effective, Not sure and ineffective in a,b and c somewhere to see in the graph not just the caption
• Figure 6b: % should be same as in the text
• Figure 7: Title should have relative to time at the end of the sentence. Also not sure why only positive graph has the dashed line.
• Table 1: include other gender categories
• Line 97 MPA network, not MPA.
• Figure 5 vs. 6: again if you remove 5 this won’t be an issue, but I don’t understand why the same method, and same graph, wasn’t used for both fisheries data and MPA opinions.
• In future, it could be useful to have specific things about MPAs that were being asked about, as well as overall opinion (e.g. performance, social impacts…)
• Line 275: This sentence seems redundant, as a positive change necessitates the others going down.
• Line 313: First sentence of discussion doesn’t mention the fisheries data opinions at all, just MPAs. These seem like they were of equal importance so would include both (since both are in the title). But again may need to be adjusted if you keep the fisheries data sections.
• Line 391 is confusing, why does the fact that they are rockfish imply the stocks are healthy?

It’s great having the time analysis, I think this is a solid conclusion from this work and a really useful point to make. Great job! At the end of the day, I think the conclusions about the MPAs are sound, but again not so sure about the fisheries data ones.

Reviewer 2 ·

Basic reporting

- The paper was well-written and the research was clearly explained and straightforward to follow.
- References provide adequate background and context for the scope of this research (i.e., collaborative fisheries research/citizen science related to CA, the CCFRP, and/or CA MPA attitudes and opinions).
- Article follows professional article structure. Nearly all tables and figures are referenced in the text (see 4. General Comments for exceptions). Raw data and analytical code was shared.
- Results are relevant, timely, and of interest, particularly to collaborative fisheries research programs and more general citizen science efforts. This research is a positive contribution to this literature.

Experimental design

- The research is a result of primary data collection utilizing an existing database (sample frame) of volunteer anglers. The research is original in its scope (i.e., collecting demographic variables and data quality and MPA attitudes from this sample frame), and fits with the “Aims and Scope” of this journal.
- The main purpose of this research, to collect demographic and attitudinal data about MPAs, fisheries data quality, and CCFRP experience from volunteer anglers, is stated clearly and is relevant to collaborative fisheries research programs and perceptions of spatial management (e.g., MPAs). The paper mentions how this study contributes to the paucity of published literature related to California-specific attitudes and perceptions related to MPAs, fisheries data quality, and the CCFRP program.
- Methods are described clearly and should be replicable, particularly given the data and R code are available in the supplemental information.

Validity of the findings

- Discussion and presentation was focused on significant results.
- Data and analytical code in R were provided and appeared to be robust.
- Conclusions were quite clear, were relevant to the original research intent, and were supported by the Tables and Figures. The paper was fairly tight (i.e., little if any extraneous narrative) and focused on the primary and significant results.
- Limited speculation is characterized with phrasing such as, “It is also possible…” (lines 398-399).

Additional comments

Line 50: Is “(California)” necessary after “1999”? “California” is mentioned multiple times in this sentence.

Line 158: What are the work experience categories? I appears to be defined for the first time in lines 219-220.

Line 202: Can you briefly characterize the 15% response rate relative to other online survey efforts found in the literature? It is unclear whether 15% is average, high, or low for an online survey in general or one that is fisheries or environmentally-focused. Perhaps survey response rates are mentioned in one of the papers already referenced? E.g., Baldassare et al., Loper, Ordoñez-Gauger et al., Rubio et al.

Lines 284-286: the order the three calculated measures of participation are listed here as year, workshops, trips but in a different order in lines 183-185 (year, trips, workshops). Small detail but I suggest using a consistent ordering throughout the paper. Also in lines 460-462 as well as in Table 2. Easiest fix would be to re-order in lines 183-185, as the rest of the paper has them ordered as year, workshops, trips.

Line 296: to aid interpretation of the statistics (e.g., RRR value) in line 297,it would be helpful to mention that “positive change in opinion” is the reference category (or referent group).

Line 356: rather than comparing the number of participants in the MLPA process to the total CA population (39 million), it may be more useful to compare to the number of coastal county residents (for example) or to the number of people living within a certain distance from the coast (e.g., 100 miles). These may be a more “fair” comparison due to their easier access to the coast and therefore to the MLPA process which, I assume, occurred primarily on the coast? However, using these other values will not change your primary point: that a relatively small proportion of Californians participated in the MLPA process. So this is definitely a suggestion but not a necessary change.

Lines 372-375: I found the beginning of this paragraph and these lines in particular very confusing. Which “two programs”? And why is it “… likely that the 111 respondents in our study represent an accurate sample of those who plan to continue with the program”? How this conclusion was reached is very unclear.

Line 388: was unsure what “accurate consensus” meant until I reached line 398. Perhaps define “accurate consensus” up front so that we understand it to mean consistency between what is believed and what actually is the state of things (if I understood it correctly!).

Line 417: missing a “)” after “Hexagrammidae)”

Lines 426-427: “i.e.” vs. “i.e.,” Throughout the paper, “i.e.” seems to be used but “i.e.,” shows up here. Choose one for consistency.

Lines 449-450: Is it possible that more conservation-minded volunteers were also more likely to respond to your survey? This gets at possible bias in your response rate, e.g., those who responded may have been more conservation-minded than those who did not respond (nonresponse bias). Is there any data on those who did not respond to your survey that could be used to test for nonresponse bias? If not, it would be worth mentioning directly in the same paragraph as the reported 15% response rate (line 202), when you characterize your response rate relative to other online survey efforts.

Lines 469-471: is it possible that volunteer anglers who participated in the Volunteer Appreciation and Data Workshops were more likely to return in subsequent years to volunteer? Did you look at this? (i.e., how participation in these workshops or average number of trips might be related to continued participation)

Lines 492-495: awkward sentence, please restructure and/or split into two sentences. E.g., “Overall, these volunteers have a positive view of the data collected for resource management and the MPAs they help to monitor. This can be attributed, in part, to long-term participation in the program.”

Lines 496-500: this is an important conclusion and contribution to this literature. Sustained volunteerism can contribute to attitude change.

Table 1 is not referenced in the text.
Articles S2 and S3 are not referenced in the text.

---

## Round 0.2 · Minor Revisions

I agree with the reviewer in that the manuscript is nearly there. If you could please respond to the reviewer comments and make changes as needed and then resubmit.

·

Basic reporting

See below

Experimental design

See below

Validity of the findings

See below

Additional comments

Thank you for taking the time to revise the manuscript and give detailed responses to the reviews. I think overall the manuscript is almost ready for publication. There are a few minor issues I think would be good to address first, but they shouldn’t take too much time.


Major comments:
My two main comments are with figure 5 and 6, which I still find confusing.

Figure 5 – Going through the questionnaire I can see why the figure is like this, but until I did that it was unclear. I still think this is misleading, and even more so than before as now if you aren’t careful you will think that the program has made huge changes in people opinions. But the reality is that half of those who had opinions prior weren’t changed, and unlike below you can’t tell if this is just because they were already positive. So if you are going to do a column I’d have it will negative change, no change and positive change. Or what would be simpler is to just provide the pie chart with four categories: no opinion before, negative change, no change and positive change.
The result here is that half of people’s opinions weren’t changed by the program, don’t try and hide this. But unlike below we can’t tell why that was.

Figure 6 – I also wonder if there is a better way to display this figure. Unless I also read the text I am left thinking that 49% of people did not change their opinion, which is true, but this is because they already had positive opinions. You should aim to let people interpret the figures without having to read the text in detail. So sort of opposite to figure 5, I wonder if here you could actually just show a pie or column of those only with a change in opinion, rather than having the no-change part in the pie chart, because unlike figure 5, they already have this information in the bars. I appreciate the difference of what is possible between these figures, so how can you make them as clear as possible?



Minor comments:
Line 139 – Would it be possible to add a few lines explaining what you mean by Margins of Error (MOE) and how this works? I can check the referenced paper, but a few lines would be good to provide an overview.

Line 154 – Smallhorn-West et al. 2019. Again though there may be better references on reflexive counterfactuals within that paper. See Franks P, Roe D, Small R, Schneider H (2014) Social assessment of protected areas: early experience and results of a participatory, rapid approach. IIED Working Paper. IIED, London

Line 155 – I think a-d are a bit of a stretch on limiting non-response bias. Sure the demographic questions do but not the first part. Anonymity equally means you can’t determine whether there is non-response bias easily. I’d just leave this as a limitation of the manuscript. No paper is perfect and this is a key caveat of yours. I think here, and also at least once in the discussion, you should explicitly state this caveat and its potential limitations, but end by saying something like ‘nevertheless, despite these limitations this study/these findings are still useful because…’

Line 166 – Again, it’s also okay to say this is a limitation. Plenty of studies use this approach, but just be clear about what it means.

Line 266 – So this clearly shows that there is in fact a bias in the population, not by demographics that you measured above, but that respondents consider themselves more conservation minded than their peers.

Line 492 – This heading is really confusing – maybe “Why volunteers may be less opinionated about fisheries data quality than MPA effectiveness” or something like that.

Lastly, as a point of clarity when reading through I’ve found that in my head it all starts becoming clear when I realize there are two things being looked at in aim c (from the introduction): fisheries data quality and opinions about MPAs. First, in the last paragraph of your introduction, make sure everything is in the same order you present it in. Second, it could help with the clarity to just restate this using similar terminology in the methods, results and discussion about i) change in opinion on fisheries data quality, and ii) change in opinions on MPAs. E.g. Line 312 heading title is confusing. Make this heading Change in opinions and then two subheadings i) fisheries data quality (not just data quality) and ii) MPAs.

---

## Round 0.3 · accepted · Accept

Apologies for the slow response on this latest revision. This manuscript is an interesting and nicely presented piece of work.